# The Relation between Ensemble Coding of Length and Orientation Does Not Depend on Spatial Attention

Melanie Kacin [1,2], Oakyoon Cha [1,3,*] and Isabel Gauthier [1]

1   Department of Psychology, Vanderbilt University, Nashville, TN 37240, USA
2   Department of Psychology, Queens College, City University of New York, Flushing, NY 11367, USA
3   Department of Psychology, Sungshin Women's University, Seoul 02844, Republic of Korea
*   Correspondence: oakyoon.cha@sungshin.ac.kr

**Abstract:** Most people are good at estimating summary statistics for different features of groups of objects. For instance, people can selectively attend to different features of a group of lines and report ensemble properties such as the mean length or mean orientation and there are reliable individual differences in such ensemble judgment abilities. Our recent study found decisive evidence in support of a correlation between the errors on mean length and mean orientation judgments ($r = 0.62$). The present study investigates one possible mechanism for this correlation. The ability to allocate spatial attention to single items varies across individuals, and in the recent study, this variability could have contributed to both judgments because the location of lines was unpredictable. Here, we replicate this prior work with arrays of lines with fully predictable spatial locations, to lower the contribution of the ability to distribute attention effectively over all items in a display. We observed a strong positive correlation between errors on the length and orientation averaging tasks ($r = 0.65$). This provides evidence against individual differences in spatial attention as a common mechanism supporting mean length and orientation judgments. The present result aligns with the growing evidence for at least one ensemble-specific ability that applies across different kinds of features and stimuli.

**Keywords:** ensemble perception; mean length; mean orientation; individual differences; spatial attention

## 1. Introduction

The visual world contains redundancies that can be summarized into ensemble percepts [1,2]. Ensemble perception is relevant to a range of situations, such as determining the average emotion of a group of faces in a crowd or perceiving the average color of the leaves on an autumn tree. In the laboratory, psychologists have asked participants to make decisions relying on summary statistics (such as the mean or variance) for a large number of features, simple or complex, under a range of experimental conditions [3–5]. One approach to the study of ensemble perception is to use individual differences across several of these tasks to learn about the structure of the abilities that may underlie these judgments.

For instance, while viewers can summarize both low- and high-level features, there is some evidence that these may rely on distinct processes. Low-level features are relatively simple and are thought to reflect the responses of specialized populations of neurons [6]. Properties such as size [1,7,8], orientation [8,9], and color [9,10], for example, have been used in ensemble perception studies. High-level features, such as the identity of an object or the expression of a face, include a more complex organization of low-level features [11]. One of the first claims about the structure of individual differences in ensemble perception is that performances in reporting the average of high-level features did not correlate with the performance reporting of the average of low-level features, while the performance was correlated within the average judgements for low-level features and within the average judgements for high-level features [9]. This landmark study had some limitations (for instance, the only high-level features were measured with face stimuli) but it cleared the

path to further investigations using individual differences to map out the structure of ensemble perception mechanisms.

Some of the work that has followed has further investigated whether ensemble perception generalizes across low-level features. Haberman et al.'s [9] idea that one mechanism supports different low-level ensemble judgments was questioned by the authors of a study [8] reporting that a performance judging the mean length and the mean orientation of ensembles of lines were not correlated; however, we performed two replications [12] with modifications of their Study 1 and found strong evidence in support of a shared variance in errors in length- and orientation-averaging, suggesting that at least one common mechanism supported the ensemble perception of these features. Although we found evidence of a shared mechanism, we did not examine what that mechanism might have been.

One possible shared mechanism driving the correlation between performance on the length- and orientation-averaging tasks would be a variability in the ability to distribute attention effectively to all lines in an ensemble. In Kacin et al. [12], the locations of the lines were randomized for each display. The ability to allocate spatial attention to single items varies in individuals based on the working memory capacity, with lower working memory capacity participants focusing their attention narrowly and higher working memory capacity participants able to flexibly distribute their attention [13,14]. Since the weights of items' contributions to ensemble judgments vary with eccentricity [15], flexibly deploying attention around the area where more items are clustered could improve ensemble judgments. Individual differences in the attention distribution ability may influence the ensemble perception ability, as attention distribution is important for ensemble perception [3,16]. Additionally, for statistical information about some low-level features to be consciously maintained, at least some attention is required. Ensemble perception of color and size was not possible, for example, when tested in an inattentional blindness paradigm and was not consciously maintained under only focal attention to specific items [17]. This suggests not only that ensemble perception is subject to attentional constraints, but that it requires a distributed rather than a focal attention [18]. Increasing the spatial predictability of ensembles would lower the contribution of individual differences in the ability to distribute attention effectively over all items in a display.

Spatial predictability has not been well-studied in ensemble perception, but visual search and change detection studies indicate a benefit from configural cues for individual items [19]. Other work found that attention is facilitated by configural and by spatiotemporal regularities when searching displays that change dynamically [20]. Summary spatial information appears to be encoded along with information about individual items, and the more easily items can be summarized, the easier observers can remember individual items [21]. Ensemble judgments may similarly benefit from spatial regularity and the ease of grouping individual items.

Low spatial predictability may increase the demands on distributed attention in ensemble perception tasks. Depending on conditions and individual abilities, there is likely to be a limit to the number of items in a display that can be sampled and that can contribute to an ensemble judgment [2,7,22,23] (but see [24]). For example, because the ensemble tasks in our previous study required adjusting attention to the new random locations of lines in each ensemble presented, participants with a high ability to distribute attentional resources could have performed better on both tasks, driving a correlation between the tasks. We note that Yoruk and Boduroglu [8] also presented lines in random locations in their Experiment 1 and did not find a correlation, suggesting that this may not be a sufficient factor to drive correlations; however, because we had made other changes to their design (e.g., avoiding lines that may be too short to support orientation perception, reducing the ambiguity due to a large range of orientations), it is possible that low spatial predictability is a necessary factor to observe variance in these tasks.

Here, we used the same length- and orientation-averaging tasks and stimuli as in the Study 1 of Kacin et al. [12], with the only exception being that in every ensemble the locations of the lines did not vary and were, therefore, completely predictable. Our goal is

to assess the replication of the observed shared variance in errors in length- and orientation-averaging in a design where individual differences in the ability to spread attention spatially should be greatly reduced, if not eliminated. We can test the extreme hypothesis that the correlation between these tasks is entirely driven by participants' having to deal with spatial unpredictability (in which case the correlation between the tasks would be 0), but our main objective, assuming there is still a correlation, is to estimate the magnitude of the correlation that cannot be driven by individual differences in the ability to allocate spatial attention. This is because in the present study, even if the participants' ability to distribute attention varied, those with a high ability would have no advantage over those with a low ability.

## 2. Materials and Methods

### 2.1. Participants

Following the methods of Kacin et al. [12], the participants were recruited using Amazon Mechanical Turk (MTurk). The participants received USD 2.50 for completing the tasks. The stopping criterion was the same as in Study 1 of Kacin et al. [12]. In the previous study, we observed a positive correlation between errors on the length- and orientation-averaging tasks. Accordingly, we estimated the support for a positive correlation relative to one of 0 ($BF_{+0}$). As in our previous study, we employed a Sequential Bayes Factor design [25] with a starting sample size of 75 participants and a stopping criterion of $BF_{+0}$ lower than 1/3 or larger than 3. After exclusions (see below), 77 participants were included in the final analyses (i.e., 30 female and 47 male) with a mean age of 40.53 years ($SD$ = 10.95).

Additionally, we used the same exclusion criterion to screen for participants who were not motivated, were not earnestly trying, misunderstood the instructions, or who experienced technical difficulty while completing the tasks. According to our previous exclusion criterion, these participants were removed from the final dataset to avoid inflating the correlations. We included participants who performed poorly but were sensitive to trial difficulty because this indicated that they were trying and should be included for an accurate estimation of the correlations. Participants who demonstrated an above-chance accuracy for the entire task were used as the basis for our trial difficulty calculation. The trial difficulty was determined by calculating the proportion of correct responses on a given trial for only those participants demonstrating an above-chance accuracy for the whole task. We included the participants if they demonstrated an above-chance accuracy on both tasks (i.e., the proportion correct > 1/5) or if their performance across the trials (with 1 for correct and 0 for incorrect) correlated positively with the trial difficulty. Six out of seventy-seven participants were included in the analyses based on their sensitivity to the trial difficulty. A positive correlation indicated that a participant performed better when the trials were easier (although the trial difficulty was randomized), suggesting that their poor performance was not a result of a lack of effort. A correlation below 0.15 was taken as insufficient evidence of effort and those participants were excluded from further analyses. We collected data from 102 participants, but excluded 25 of them according to our pre-defined criteria. All the procedures were approved by the Vanderbilt University Institutional Review Board in charge of overseeing the protection of human subjects. Informed consent was obtained prior to the experiment.

### 2.2. Materials

As in Kacin et al. [12], the ensembles used in both tasks consisted of 12 lines of varying lengths and orientations. The lengths of the lines increased in increments of 4 pixels from 44 to 116 pixels for a total of 19 lengths. We reported the stimulus lengths in pixels because the degrees of visual angle could differ depending on a participant's monitor size and screen resolution. The orientations of the lines increased in 4.5° increments clockwise from the vertical meridian from 4.5° to 85.5° for a total of 19 orientations.

The lines used in the length-averaging task could have had a mean length of 56, 68, 80, 92, or 104 pixels, with a standard deviation (SD) of 8 or 12 pixels. The lines could

have had a mean orientation of 31.5°, 45°, or 58.5° with a SD of 9° or 13.5°; therefore, using all the possible combinations of mean lengths, length SDs, mean orientations, and orientation SDs, we produced 120 unique ensembles (i.e., 5 mean lengths × 2 length SDs × 3 mean orientations × 2 orientation SDs × 2 repetitions). A display of five lines (56, 68, 80, 92, and 104 pixels long) at a 45° angle was shown during the response screen, during which the participants selected the mean length of each ensemble.

The lines used in the orientation-averaging task could have had a mean orientation of 18°, 31.5°, 45°, 58.5°, and 72°, with a SD of 9° or 13.5°. The lines could have had a mean length of 68, 80, or 92 pixels with a SD of 8 or 12 pixels; therefore, using all the possible combinations of mean orientations, orientation SDs, mean lengths, and length SDs, we produced 120 unique ensembles i.e., (5 mean orientations × 2 orientation SDs × 3 mean lengths × 2 length SDs × 2 repetitions). A display of five 80-pixel lines (18°, 31.5°, 45°, 58.5°, and 72°) was shown during the response screen, during which the participants selected the mean orientation of each ensemble.

There were 5 counterbalanced versions of the experiment, with each version associated with one fixed spatial configuration for the length-averaging task and another fixed spatial configuration for the orientation-averaging task (Figure 1). The locations of the lines (i.e., the spatial configuration) for the length-averaging task were selected from the 5 trials from our previous Study 1 [12] with the highest mean accuracy from the length-averaging task; the locations of lines for the orientation-averaging task were selected in the same way. Then, two spatial configurations, one for the length- and the other for the orientation-averaging task, were paired randomly, resulting in the total of 5 pairs (i.e., 5 versions of the experiment). In all versions of the experiment, the lengths and orientations of the lines in each trial were kept the same and the trials were administered in a single, fixed random order. The only difference in the 5 versions was the locations of the lines, which were distributed across a five-by-four grid of possible locations and were not randomly jittered as in our previous study. This was done to further remove the need to reallocate attention for random jitters in each new display. For each possible distribution of locations in the grid, 3 lines were placed in each of the 4 rows.

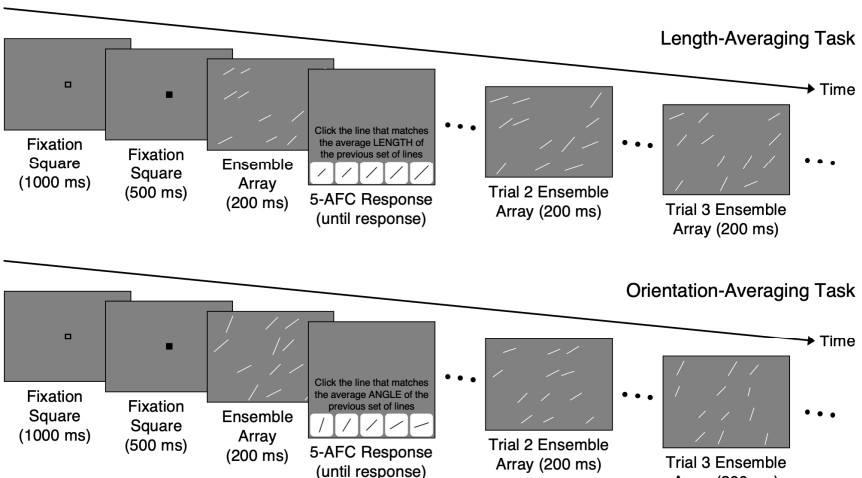

**Figure 1.** Example of the sequence of one trial and examples of ensemble arrays in the two subsequent trials in the length-averaging task (**top**) and in the orientation-averaging tasks (**bottom**). The size of fixation squares and the text in the response displays are not shown to exact scale for presentation purposes. The words in the response display read "Click the line that matches the average LENGTH/ORIENTATION of the previous set of lines."

### 2.3. Procedure

The participants were randomly assigned to one version of the study that contained both the length- and orientation-averaging tasks. The length-averaging task preceded the

orientation-averaging task in all versions to avoid introducing another source of variability. For each trial in both tasks, the participants judged the average length or orientation of the lines in an ensemble (Figure 1). To prevent the participants from processing each line individually, the presentation time for each ensemble was 200 ms. A fixation square sequence lasting 1500 ms (i.e., a 1000 ms open square and 500 ms black square) preceded each ensemble. The participants were then prompted to estimate the average length or orientation of the ensemble by clicking on one of the 5 buttons, each depicting a line having a specific length or orientation. There was no time limit to make the average length/average orientation judgments. The participants completed 10 practice trials with non-guided feedback (i.e., the feedback told the participants whether they were correct or incorrect but did not indicate the correct answer). The practice trials allowed participants to familiarize themselves with the procedure and provided guidance on how to compute the average length or orientation of an ensemble. In the practice trials, the mean length or orientation of the ensembles was one of the 5 values used in the main tasks, but the SD was 4 pixels or 4.5°. Additionally, the lines in each practice ensemble were only placed in columns 1, 3, and 5 in the grid. This resulted in ensembles with lines that were easier to average compared to the main task ensembles. Each participant in all versions completed 120 trials of the length-averaging task followed by 120 trials of the orientation-averaging task.

As in our previous study (but different from Yoruk and Boduroglu [8], who used a continuous adjustment method), we used a five-alternative forced choice (5-AFC) task to record the responses. Because with this method the responses are limited to 5 options, the sensitivity of error measures relative to a continuous adjustment method may be reduced, but we have previously demonstrated that the precision is more than sufficient to detect a correlation [12]. We used 5 choices as a compromise between reducing the chance level and allowing for rapid decisions with less deliberation. This method was also compatible with our in-house experimental platform.

### 2.4. Analysis

Bayesian correlation analyses were conducted using JASP (version 0.16.3; JASP team, Amsterdam, The Netherlands) [26]. The Bayes factor (BF) is an indicator of the relative evidence for two competing models. We report $BF_{+0}$, which demonstrates support in favor of a positive correlation (which would indicate a common mechanism underlying the length and orientation judgments) relative to a correlation of 0. Jeffreys' criteria [27] was also used to interpret the BF values in support of a positive correlation, with values between 1 and 3 indicating very weak support, values between 3 and 10 indicating substantial support, values between 10 and 30 indicating strong support, values between 30 and 100 indicating very strong support, and those above 100 indicating decisive support.

The correlation between the mean absolute error on the length- and orientation-averaging tasks was calculated by analyzing all participants who completed each version of the tasks. We used a 5-AFC task, with each alternative choice corresponding to a specific length or orientation. The error (distance) between the true mean length or orientation and the chosen length or orientation for each participant was first calculated. Then, we computed the correlation between errors in the two tasks using the average of absolute errors across participants.

### 3. Results

The average and SD of the accuracy was $M = 0.43$ and $SD = 0.11$ for the length-averaging task, and $M = 0.50$ and $SD = 0.16$ for the orientation-averaging task. The average and SD of the absolute error was $M = 8.93$ pixels and $SD = 2.90$ pixels for the length-averaging task, and $M = 9.23°$ and $SD = 4.67°$ for the orientation-averaging task. The reliabilities of the measurements for the length- and orientation-averaging task were $\alpha = 0.93$ and $\alpha = 0.96$, respectively. We found decisive evidence for a positive correlation between the length and orientation averaging errors compared to a null correlation ($r = 0.65$, 95% CI = (0.49, 0.76), and $BF_{+0} = 1.89 \times 10^8$; Figure 2). We repeated the correlation

analysis with 71 participants who performed above chance in both the tasks, and found a very similar result ($r = 0.69$, 95% CI = (0.54, 0.79), and $BF_{+0} = 8.61 \times 10^8$). Since the reliabilities of the two measures posed a limitation on the maximum correlation that could be observed, we estimated a disattenuated correlation by dividing the observed correlation by the square root of the product of the two measures' reliabilities [28]. The disattenuated correlation between the length- and orientation-averaging errors was $r = 0.69$, suggesting a 47.6% shared variance between the two ensemble tasks. One may be concerned that the absolute error measure is confounded with biases. For instance, longer lines may be overweighted during averaging due to their saliency [29]. In that case, the mean length would be overestimated whereas there would be no such biases in the mean orientation judgments. Biases in one task, however, could have only reduced the correlation. We calculated the standard deviation of errors for the mean length and mean orientation judgments and found a larger correlation between the standard deviation measures ($r = 0.72$, 95% CI = (0.59, 0.81), and $BF_{+0} = 2.57 \times 10^{11}$).

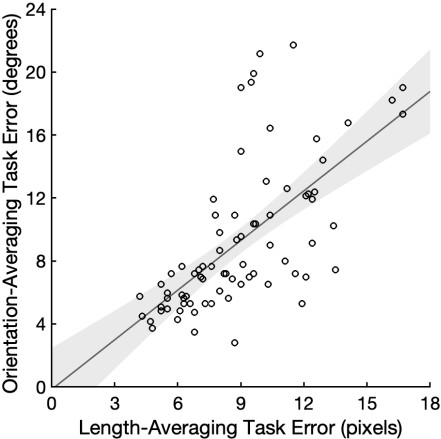

**Figure 2.** Scatterplot demonstrating the relationship between length-averaging errors and orientation-averaging errors, with each data point representing a unique participant. The linear best fit is shown (95% CI).

The correlation we found in the current study is clearly no less than the correlations (e.g., Study 1: $r = 0.62$, 95% CI = (0.45, 0.73); Study 2: $r = 0.63$, 95% CI = (0.46, 0.74)) that we observed in Kacin et al. [12], suggesting that an ability to allocate attention flexibly in each trial is unlikely to be responsible for the common variance between the length and orientation average judgments. Either our subjects did not vary in their ability to allocate spatial attention (which appears unlikely), or this is not an important contribution to individual differences in these tasks.

## 4. Discussion

We replicated Kacin et al.'s [12] result of a correlation for errors made by participants when reporting the average length or the average orientation of arrays of 12 lines. At this point, the correlation between the errors for these two judgments proved to be quite similar across three replications. We found this correlation to vary between $r = 0.62$ and $r = 0.69$ in three different designs. In Study 1 of Kacin et al. [12], this effect was obtained when the participants estimated the average length or orientation of the arrays of 12 lines that varied on both features. That is, the participants were asked to attend to one or the other feature in different blocks. In Study 2 of Kacin et al. [12], we only varied the task-relevant feature for each type of judgment and obtained virtually the same correlation. This ruled out the possibility that the correlation arose from participants being unable to ignore variability in the task-irrelevant dimension. In the present work, we again observed the same magnitude of a correlation between the two kinds of judgments, even though we removed any spatial uncertainty from our stimuli. By removing spatial unpredictability, the participants who



were better able to adjust to the flexible arrangements in the previous studies would have had less of an advantage in this present study relative to other participants with a lower ability to adjust their attention. This rules out the possibility that individual differences in the ability to direct spatial attention is the mechanism responsible for these correlations. The stability of this correlation across these different conditions leads us to conclude that at least one common mechanism is at least partially responsible for the averaging of these two features. This is consistent with Haberman et al.'s [9] proposal for a low-level ensemble processor that introduces a shared source of noise in these judgments.

The replicability of this correlation comes in contrast with work by Yoruk and Bodoruglu [8]. In three experiments, these authors included conditions in which the participants also made separate judgments for the length or orientation of 12 lines, varying the duration of the display (from 50 ms to 200 ms), whether it was masked or not. There was little effect of the presentation duration or masking on errors. The largest correlation they obtained between the errors for the two tasks was 0.28, with many of their results providing Bayes factors that either supported a null result over a correlation, or were inconclusive. Aside from the difference in power between their three studies (i.e., $N$s = 21, 24 and 37) vs. the three studies in which we found a correlation (i.e., $N$s = 81, 75 and 77), the most salient difference between the two sets appears to be in the precision of the length judgments (i.e., the precision of judgments was very similar across their studies; therefore, we here refer to the precision in their Experiment 1). The precision of the length judgments in their Experiment 1 was 19.6 pixels with an SD of 7.76. In contrast, the data we collected in this study and in Kacin et al. [12] resulted in length judgments ranging from 8.52 to 9.48 pixels, with SDs ranging from 2.4 to 3.84. Not only were Yoruk and Bodoruglu's [8] participants' length judgments less precise, but those judgments were also much more variable than those made by our participants. We note that the smaller variability across the participants in our measures did not limit the reliability of the individual differences we measured (with $\alpha > 0.9$ in the present work). Yoruk and Bodoruglu [8] did not report the reliability of their measurements.

We surmise that this difference in precision (even when the number of lines and duration is the same across sets of studies) stems from the fact that Yoruk and Bodoruglu [8] included some very short lines (as short as 24 pixels, compared to our shortest length of 44 pixels). Their participants may have sometimes failed to encode outlier short lines and even more problematically, they may have varied in their sensitivity to these short lines. This would confound the errors for length judgments with a different source of individual differences. The effect of missing short lines would not be the same for orientation judgments, as the orientation and length were not correlated in the stimuli. Missing short lines would lead one to underestimate the average length, but only result in an orientation estimate based on a smaller number of items with the same mean orientation. Yoruk and Bodoruglu [8] drew additional conclusions in their studies from other conditions in which the participants performed the two judgments within the same trials. The relation between the tasks was, however, the same across their single or mixed tasks conditions (with little to no correlation); therefore, we suggest that the same considerations apply to conclusions about mixed tasks trials.

Our results provide further evidence against independent statistical summary mechanisms for ensemble coding of these two low-level features, by ruling out a plausible alternative source of individual differences. Could using identical arrays of lines across the two tasks allow for other explanations for the correlation we observed, aside from those related to the ensemble coding proper? We do not think so, for a few reasons. First, note that Study 2 in Kacin et al. [12] used different types of displays for the two tasks (varying only in one or the other feature), without any impact on the correlation observed. Second, Haberman et al. [9] found correlated errors for estimating the mean orientation for different stimuli (e.g., Gabors vs. triangles), as well as for estimating the mean orientation and mean color for different stimuli. Third, in a related line of studies, we found that estimating the mean and diversity of the size of circle arrays, in two different ensemble coding tasks, led

to correlated errors, even after controlling for individual differences in the error for judging the size of individual circles [30]. Finally, in the realm of higher-level ensemble coding, estimating the average identity of complex objects such as cars, birds and planes leads to correlated errors, even after controlling for individual differences in the ability to recognize individual objects within these categories [31]. This was also confirmed in a large structural equation modeling study ($n = 284$) in which average identity judgments for six categories of complex objects (either familiar or novel) loaded strongly on an ensemble coding factor that shared about 42% of the variance with a domain-general object recognition ability [32]. Importantly, this left 58% of the shared variance across ensemble judgments unexplained by the object recognition ability (or a measurement error), pointing to a source of shared noise across the ensemble judgments for displays of objects that have very little in common.

We acknowledge some limitations of the present work. First, some participants may have relied on strategies other than averaging to complete the tasks, such as randomly sampling one line and choosing the probe that most closely matched the length or orientation of that item. Our task was not one in which we sought to demonstrate that performance can only be explained by the distributed-attention processing of multiple items, as many other studies have (e.g., [18,33]). Rather, our instructions encouraged the distributed attention and processing of as many items as possible while the participants may have chosen to use an approach with narrower attention. In that sense, we may have, in part, been measuring individual differences in the adoption of an ensemble judgment strategy, rather than a direct measure of its precision. Second, what motivated this work was the desire to assess whether the spatial unpredictability in our prior work may have been an important (or even the sole) source of correlation across feature judgments. This work did not manipulate the magnitude of spatial predictability, and while the present design represents the absence of unpredictability, it should not be considered maximal by any means in our prior work. Future work could use a larger manipulation of spatial unpredictability to test for the possibility that it could influence the performance and shared variance to some extent, even if this is small.

It is notable that the growing literature on individual differences in ensemble perception with both high- and low-level features converges in finding similar patterns of shared variance across different tasks. Whether the initial division of abilities for the high- and low-level ensemble coding proposed by Haberman et al. [9] holds up to the test of further studies, it is perhaps too soon to know. Haberman et al. only used faces (i.e., an identity vs. an expression judgment) to measure the higher-level ensemble coding abilities. No study has yet measured the correlation between errors in the ensemble coding for faces and non-face objects, or between ensemble judgments with low-level features and complex non-face objects. Haberman et al. [9] compared two extreme process models for ensemble perception, the first in which a domain-general ensemble processor introduced noise uniformly to all ensemble processes, and the second where separate low- and high-level ensemble processors each built on the processes responsible for processing low- and high-level individual features. They provided relatively strong evidence against a fully domain-general ensemble processor that would include ensemble coding for faces, but further work with non-face objects and low-level features is needed to rule out intermediate models where high-level non-face features might group with low-level features. Alternatively, it is also possible that the independence of face and low-level coding judgments does not replicate in future studies with a larger range of low-level tasks. Even in studies measuring individual differences for individual item processing, the processing of faces often still shows at least a small degree of overlap with that of non-face objects [34,35].

## 5. Conclusions

We replicated prior work by Kacin et al. [12] that found a robust correlation between the errors for judgments of the mean size or mean orientation of the arrays of lines. To rule out the contribution of individual differences in the ability to distribute attention effectively over all items in a display, we used arrays of lines with fully predictable spatial

locations. Bayesian analyses indicated decisive support for a positive correlation between errors on the length- and orientation-averaging tasks compared to a null correlation ($r = 0.65$). These results rule out individual differences in spatial attention as an important source of the common mechanism supporting mean length and orientation abilities. The present result aligns with the growing evidence for at least one ensemble coding-specific ability that applies across different kinds of features and stimuli. The implication for the use of individual differences to understand the relationship of ensemble perception to other abilities is important. To the extent that individual differences are unique to different features or tasks, we could not define a latent ability that contributes to different ensemble perception tasks (as suggested by Haberman et al. [9]), and we could not investigate how this ability relates to other constructs.

**Author Contributions:** Conceptualization, M.K., O.C. and I.G.; methodology, M.K., O.C. and I.G.; software, M.K.; validation, M.K., O.C. and I.G.; formal analysis, M.K.; investigation, M.K.; resources, I.G.; data curation, O.C.; writing—original draft preparation, M.K. and I.G.; writing—review and editing, M.K., O.C. and I.G.; visualization, O.C.; supervision, I.G.; project administration, I.G.; funding acquisition, I.G. All authors have read and agreed to the published version of the manuscript.

**Funding:** This work was supported by the NSF (SMA-1640681) and by the David K. Wilson Chair Research Fund (Vanderbilt University).

**Institutional Review Board Statement:** The study was conducted in accordance with the Declaration of Helsinki, and approved by the Institutional Review Board of Vanderbilt University (protocol number: 050082, date of approval: 29 April 2021).

**Informed Consent Statement:** Informed consent was obtained from all subjects involved in the study.

**Data Availability Statement:** Data reported in this study are available at https://osf.io/hcrg8 (accessed on 30 October 2022).

**Conflicts of Interest:** The authors declare no conflict of interest.

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
