# Peer review of "The Relation between Ensemble Coding of Length and Orientation Does Not Depend on Spatial Attention"

_2411-5150, 2022_

Round 1

Reviewer 1 Report

Review of vision-2031937, “The relation between ensemble coding of length and orientation does not depend on spatial attention” by M. Kacin, O. Cha, and I. Gauthier.

In much the same way as researchers have questioned whether face recognition is mediated by domain-general or highly-specialized neural processes, the question of whether ensemble coding is carried out by domain-general or domain-specific mechanisms has also received a fair bit of attention. One approach to addressing this is to ask whether performance on different types of tasks requiring the computation of summary statistics is correlated. The findings have been mixed, with some investigations suggesting strong correlations between otherwise dissimilar tasks and others finding minimal correspondence in performance. The purpose of the present work is to eliminate a possible confounding variable in previous studies, differences in demands on the deployment of spatial attention.

To compute summary statistics in a visual display where the items are segregated by space rather than by time, attention must be diffusely deployed to the display. However, the authors note that people who are low in working memory (WM) capacity allocate spatial attention more narrowly than those high in WM capacity, which should be detrimental to ensemble coding. Thus, individual differences in the ability to efficiently allocate spatial attention, rather than individual differences in ensemble coding per se, could account for any shared variance in performance observed between a pair of ensemble tasks. Previous studies yielding significant correlations in performance used displays in which the spatial positions of the items was randomized, introducing spatial uncertainty into the tasks. In the present study, the authors eliminate the spatial uncertainty of the items comprising the ensembles. Subjects viewed groups of 12 lines differing in either length or orientation and were instructed to choose which of five probes corresponded to the average length or orientation (blocked) of the ensemble shown in the preceding display. Removing spatial uncertainty from the task should reduce the contribution of individual differences in deployment. If this is a source of shared variance in performance across the two ensemble coding tasks, correlations should be much weaker or eliminated in the present study.

The results yielded significant correlations in performance across the two types of judgments, suggesting that 1) there is some shared underlying mechanism that supports performance in the two types of ensemble judgments, but that 2) it is not the efficiency with which attention can be deployed to the items in the displays.

Evaluation

1)    I wasn’t surprised to find that eliminating spatial uncertainty did not reduce the correlation observed in previous studies because this manipulation would not be expected to change how attention needs to be allocated to the displays to perform well. Presumably, a diffuse deployment of attention is needed to perform well in ensemble judgments irrespective of whether observers know where the items will be or not. Assuming that items are distributed broadly across the entire display (rather than, say, clustering in one quadrant) as seems to have been the case both here and in the Kacin et al. (2021) paper, I’m not sure why the authors would think that it matters whether the locations are predictable or not. Either way, a diffuse deployment of attention is required, so the present study isn’t really different from the 2021 studies.

2)    I also question whether subjects need to compute ensemble statistics at all in this task, especially given the predictable nature of the locations of stimuli. I did a quick calculation of what performance would be expected to look like if observers simply chose a random item (made easier by the certainty in their locations, and made more likely by the short exposure duration of the display) and chose the probe that most closely matched the length of the randomly selected item. Taking length as an example, ensembles with a mean length of 56 pixels are made up of lines drawn from the set 44, 48, 52, 56, 60, 64, and 68 pixels. Randomly sampling one of these and choosing the probe that most closely matches the length of the sampled item (56, 68, 80, 92, or 104 pixels) means that the 56 pixel probe will be chosen if 44, 48, 52, or 60 are sampled. This will lead to 0 error 100% of the time for 44, 48, and 52, since these items only ever appear in the ensemble with a mean of 56 pixels. Sampling 56 or 60 and choosing the 56 pixel probe will be correct half the time, since these items can appear in ensembles with means of either 56 or 68 pixels, and lead to error of 12 pixels the other half of the time (when 56 is chosen but in fact the sampled item came from an ensemble with a mean of 68 pixels). Similarly, sampling 64 should lead subjects to choose the 68 pixel probe (since it is closest), which would be correct half the time and yield an error of 12 pixels the other half of the time. A line of length 68 can appear in ensembles with a mean of 56, 68, or 80, meaning that when the 68 pixel probe is chosen, this is correct 1/3 of the time and leads to an error of 12 pixels 2/3 of the time.

By extending this logic to the entire set (and, analogously to the set of orientations, since the arrangement is the same, only the step size differs) and then computing the average error across the entire set, we would expect subjects to be able to achieve an average error of 5.49 pixels or 6.17 degrees, which is actually less than what was observed in the present study. That is, an ideal observer who does not try to compute ensemble statistics but simply samples one line and chooses the probe that best matches it could achieve better precision that what was actually observed in the present study. Thus, good performance could be observed in these tasks even if observers aren’t computing summary statistics at all. We can also use this approach to estimate accuracy (how often the correct probe is chosen). This ideal observer would choose the correct probe on approximately 54% of trials – far exceeding chance (20%).

Given the authors are hoping to draw conclusions about shared mechanisms in different types of ensemble judgments, I think it is critical to use a task that only allows participants to succeed if they are actually computing averages.

3)    If observers are actually attending to only a single line, they only need to use focused attention, which the authors note would be used by those low in WM capacity anyway. So the present study might not be the best test of whether individual differences in the ability to deploy spatial attention accounts for shared variance in this task since both low and high VM capacity subjects could be successful in the task using focused attention to a single line.

Minor comments

Line 44: “…while performance within each level was correlated.” I wasn’t clear on what was correlated within each level – were two different dependent variables measured in this study?

Line 65: “Individual differences in attention distribution ability may influence ensemble perception ability, as attention distribution is important for ensemble perception.” It would be helpful to clarify that it’s specifically the ability to diffuse spatial attention across an entire display that is important for ensemble perception.

Line 77: I found the sudden shift from discussing configural and spatiotemporal regularities to modelling working memory capacity in change detection very jarring. I think this section would read better if this sentence was removed.

Line 121: “Trial difficulty was calculated using the proportion of correct responses on a given trial…” This confused me because there is only one response per participant on a given trial. I’m guessing the authors mean that they determined the percentage of participants who got the correct answer to a specific display (trial) in the experiment and used that as an index of how difficult that trial was, but this should be clarified. I thought the calculation of the correlation between trial difficulty and accuracy as a means of determining whether an observer was trying (i.e., was sensitive to trial difficulty) was very clever – I haven’t seen this before.

Line 127: “A correlation below .15 was taken as insufficient evidence of effort”. This seems like a really low bar to hit. I realize that any cut-off is arbitrary and something has to be chosen, but it might be good to try a few different cut-offs and see if it makes much of a difference because I think it’s hard to make the case that a correlation of, say, .20 shows that people were actually sensitive to trial difficulty. How many subjects would be removed if, say, a correlation of .50 or even .75 were used?

Line 180: Why wasn’t the order of presentation of the length averaging and orientation averaging blocks counterbalanced? I suspect this was intentional (to avoid introducing another source of variability between the two tasks) but this should be explained.

Results: What was overall accuracy (percentage of trials on which the correct probe was chosen) for the length judgment and orientation tasks? I think this should be reported. As noted above, sampling a single item and choosing the closest probe should yield around 54% correct responses (approximately; the actual number depends on how frequently each item actually appears in a set but I didn’t simulate this because it doesn’t change the average error expected). If observers are doing much better than this, this might be evidence that they are sampling more than a single item and choosing the correct probe more often as a result.

Line 249: “Either our subjects did not vary in the ability to allocate spatial attention (which appears unlikely), or this is not an important contribution to individual differences in these tasks.” I agree that it’s unlikely that subjects didn’t differ in the ability to allocate spatial attention. However, for the reasons I outlined above, I don’t believe the present tasks would tap into these individual differences, assuming high WM capacity subjects can allocate attention to an individual line as well as those low in WM capacity.

Line 336: “…a domain-general ensemble processor introduces noises uniformly…” I think “noises” should just be “noise.”

I hope the authors find these comments helpful.

Reviewer 2 Report

The manuscript covers an interesting topic and is generally well presented. I will make a few minor suggestions that I consider could improve the quality of the paper.

The introduction is quite complete, although I would suggest setting out in more detail the main hypotheses of the study along with the objectives of the study.

I recommend including in the "participants" section the socio-demographic characteristics of the sample (gender, age, etc).

I also consider it necessary to include a sub-section within the methodology where the ethical aspects of the research are presented.

I recommend including a subsection in the discussion in which the limitations of the study are presented.

And finally, I suggest including the practical implications and theoretical contributions of this research in detail in the conclusion section.
